# Significance of Identifying Key Genes Involved in HBV-Related Hepatocellular Carcinoma for Primary Care Surveillance of Patients with Cirrhosis

**DOI:** 10.3390/genes13122331

**Published:** 2022-12-10

**Authors:** Yaqun Li, Jianhua Li, Tianye He, Yun Song, Jian Wu, Bin Wang

**Affiliations:** 1Department of Pharmacy, Huashan Hospital, Fudan University, Shanghai 200040, China; 2Department of General Surgery, Huashan Hospital, Fudan University, Shanghai 200040, China; 3National Clinical Research Center for Aging and Medicine, Huashan Hospital, Fudan University, Shanghai 200040, China

**Keywords:** hepatocellular carcinoma (HCC), co-immunoprecipitation/mass spectrometry (CO-IP/MS), HBV-encoded oncogene X protein (HBx), bioinformatic analysis, hepatitis B virus (HBV)

## Abstract

Cirrhosis is frequently the final stage of disease preceding the development of hepatocellular carcinoma (HCC) and is one of the risk factors for HCC. Preventive surveillance for early HCC in patients with cirrhosis is advantageous for achieving early HCC prevention and diagnosis, thereby enhancing patient prognosis and reducing mortality. However, there is no highly sensitive diagnostic marker for the clinical surveillance of HCC in patients with cirrhosis, which significantly restricts its use in primary care for HCC. To increase the accuracy of illness diagnosis, the study of the effective and sensitive genetic biomarkers involved in HCC incidence is crucial. In this study, a set of 120 significantly differentially expressed genes (DEGs) was identified in the GSE121248 dataset. A protein–protein interaction (PPI) network was constructed among the DEGs, and Cytoscape was used to extract hub genes from the network. In TCGA database, the expression levels, correlation analysis, and predictive performance of hub genes were validated. In total, 15 hub genes showed increased expression, and their positive correlation ranged from 0.80 to 0.90, suggesting they may be involved in the same signaling pathway governing HBV-related HCC. The GSE10143, GSE25097, GSE54236, and GSE17548 datasets were used to investigate the expression pattern of these hub genes in the progression from cirrhosis to HCC. Using Cox regression analysis, a prediction model was then developed. The ROC curves, DCA, and calibration analysis demonstrated the superior disease prediction accuracy of this model. In addition, using proteomic analysis, we investigated whether these key hub genes interact with the HBV-encoded oncogene X protein (HBx), the oncogenic protein in HCC. We constructed stable HBx-expressing LO2-HBx and Huh-7-HBx cell lines. Co-immunoprecipitation coupled with mass spectrometry (Co-IP/MS) results demonstrated that CDK1, RRM2, ANLN, and HMMR interacted specifically with HBx in both cell models. Importantly, we investigated 15 potential key genes (*CCNB1*, *CDK1*, *BUB1B*, *ECT2*, *RACGAP1*, *ANLN*, *PBK*, *TOP2A*, *ASPM*, *RRM2*, *NEK2*, *PRC1*, *SPP1*, *HMMR*, and *DTL*) participating in the transformation process of HBV infection to HCC, of which 4 hub genes (*CDK1*, *RRM2*, *ANLN*, and *HMMR*) probably serve as potential oncogenic HBx downstream target molecules. All these findings of our study provided valuable research direction for the diagnostic gene detection of HBV-related HCC in primary care surveillance for HCC in patients with cirrhosis.

## 1. Introduction

The vast majority of HCC cases result from persistent liver inflammation and cirrhosis. Cirrhosis is the final stage of the disease before liver cancer develops and is one of the risk factors for the development of liver cancer. The incidence of HCC is much higher in the cirrhotic state than in the non-cirrhotic state [1,2]. However, there are significant obstacles to primary care surveillance for HCC in patients with cirrhosis. Little is known about practice patterns for HCC primary care surveillance, with less than 20% of patients with cirrhosis undergoing HCC testing every 6 months. In addition, primary care testing for HCC is confounded, with many primary care providers (PCPs) erroneously believing that the clinical examination of α-fetoprotein (AFP) expression is an effective surveillance tool. At a threshold of 20 ng/mL, the sensitivity of AFP for detecting HCC smaller than 5 cm ranges from 49% to 71%, while the specificity ranges from 49% to 86%. In addition, AFP does not clearly differentiate between HCC and cirrhosis, which severely limits its utility for guiding individualized diagnostic and therapeutic strategies [3]. Moreover, modern medical research demonstrates that the liver has no pain sensation, and even if liver disease has begun, the body is unable to perceive it through pain feedback mechanisms. During the transformation of cirrhosis to liver cancer, there are no obvious clinical symptoms, causing many patients to be in the middle and late stages of the disease when liver cancer is diagnosed, resulting in an exceedingly poor prognosis and increased patient mortality [4,5]. The presence of both inflammation and cirrhosis complicates the early detection of HCC, and few HCC biomarkers have adequate diagnostic sensitivity for HCC in its early stages. Even though patients are routinely monitored for HCC, diagnosis will be missed because of the poor accuracy and sensitivity of surveillance methods [6].

However, the process by which cirrhosis transforms into liver cancer is still unknown. Current research on the pathogenesis of HCC focuses on identifying biomolecules involved in cancer progression and tumor cell response, particularly those specifically activated by HBx. HBx, the principal oncogenic protein of HBV, may perform a crucial function in hepatocarcinogenesis. Understanding the HBx-mediated driver complex aspects of HCC, particularly HBx-regulated downstream genes, would facilitate the search for biomarkers from a carcinogenic perspective and construct an effective link between biomarkers and the disease process; therefore, assigning “stage” symbolic attributes to markers offers a novel strategy for the effective prevention, diagnosis, and treatment of hepatocarcinogenesis in patients with cirrhosis [7].

In this study, we identified genes uniquely expressed in HBV-related HCC and confirmed a trend of increased expression of these hub genes during the progression of the disease from cirrhosis to HCC, laying the groundwork for future research into the early warning diagnosis of HCC. Next, we assessed the clinical significance of these hub genes in HCC. Lastly, we investigated the interaction of these hub genes with HBx, using the Co-IP/MS technology based on proteomics to confirm the oncogenic significance of these key molecules in hepatocarcinogenesis. 

## 2. Materials and Methods

### 2.1. Acquisition of Datasets and Candidate DEG Screening

This study used microarray data from the GEO database (https://commonfund.nih.gov/GTEx/; accessed on 4 September 2022). Only clinical tissue-related records were included, and datasets pertaining to mRNA expression at the animal or cell line experiment level were excluded. The data for mRNA expression were taken from GSE121248 (70 HBV-infected HCC tissues and 37 para-cancer liver tissues) [8]. Two methods of DEG screening were used. GEO2R (https://www.ncbi.nlm.nih.gov/geo/geo2r/; accessed on 4 September 2022), a GEO database analytic tool, was initially used to screen DEGs from GSE121248. Next, the R-limma package in R software was used to screen and validate the DEGs in GSE121248. The data were normalized using the Normalize Between Array function from the Bioconductor project. The “*p* value < 0.01 and |log_2_FC|>2” phrase was used as the cut-off criterion in the two methods. Using the R-ggplot2 package, Veen diagrams and a volcano plot were developed to better illustrate overlapping DEGs (candidate DEGs) that were identified by two methods. The R-ComplexHeatmap package was used to generate heatmaps for candidate DEGs. For non-linear dimensionality reduction, the R-umap and R-ggplot2 packages were used to execute UMAP, the nonlinear dimensionality reduction algorithm. PCA was performed using the R-Consensus Cluster Plus package. All study procedures were conducted in accordance with the protocol. The approval and informed consent of a local ethics committee were not required for the analysis of data from a public database.

### 2.2. Function and Pathway Enrichment for DEGs 

Gene Ontology (GO) analysis and the Kyoto Encyclopedia of Genes and Genomes (KEGG) analysis were used to evaluate candidate DEGs with the R-clusterProfiler package to ponder possible biological functions and signals [9]. GO analysis is a commonly used bioinformatics technique for annotating and analyzing the biological functions of genes, which include biological process (BP), molecular function (MF), and cell composition (CC). KEGG is a valuable database for implementing advanced functions and applications of biological systems based on large-scale molecular mapping [10]. The main enrichment functions and pathways of DEGs were screened with a threshold of adjusted *p* < 0.05. Significant GO and KEGG items and pathways were visualized using the R-Goplot package.

### 2.3. Establishment of a PPI Network and Identification of Hub Genes

To investigate the physical and functional associations of candidate DEGs, a PPI network was constructed using the Search Tool for the Retrieval of Interacting Genes/Proteins database (STRING11.0) [11]. The regulatory network was represented using Cytoscape (version 3.6.1). Using CytoHubba [12], a Cytoscape plug-in, nodes in a loaded network were ranked based on their characteristics using multiple topological algorithms in order to mine hub genes from the PPI network. The top 20 nodes (hub genes) were selected using the Maximal Clique Centrality (MCC) method, which predicts essential proteins with greater accuracy [13].

### 2.4. Validation of Hub Gene Expression Level in Cirrhosis and HCC Tissues

The identification of hub genes in HCC were verified using The Cancer Genome Atlas (TCGA, https://portal.gdc.com; accessed on 6 September 2022) database, and the analysis data came from LIHC Level3 HTSEQ-FPKM RNA-seq [14]. The two-gene correlation map was realized using the R-ggstatsplot package. Spearman correlation was used to describe the correlation relationship [15]. The expression trend of hub genes between cirrhosis and HCC was explored in GSE10143 (225 hepatitis cirrhosis tissues and 80 HCC tissues) [16], GSE54236 (80 HCC tissues and liver cirrhosis tissues from the same patients), GSE25097 (222 HCC tissues and para-cancer tissues, 40 cirrhosis tissues, and 6 healthy liver tissues), and GSE17548 (13 HBV-related cirrhosis tissues and 10 HBV-related HCC tissues) [17] datasets from the GEO database. The R-limma and R-edgeR packages were used to analyze the expression level of the hub genes with threshold parameters |log_2_FC| > 1 and FDR < 0.05 [18]. The expression profile was then visualized using the R-ComplexHeatmap (version 3.3.3) package.

### 2.5. Multifaceted Clinical Predictive Performance Analysis of Hub Genes for HCC

Kaplan-Meier analysis was conducted to compare the overall survival (OS) rate between the high- and low-hub-gene-expression groups using the *p*-value determined by the log-rank test. Hazard ratios (HRs) accompanied by 95% confidence intervals (CIs) and log-rank *p*-values were calculated and plotted [19]. We constructed a prediction formula as follows: risk score = (β1 × gene 1 expression level) + (β2 × gene 2 expression level) + ... + (βn × gene n expression level). The formula is a linear combination of the gene expression value of each gene and the regression coefficient (β) [20]. The R-survminer, R-survival, and R-ggrisk packages were used to draw a risk plot [21,22]. The objective of constructing a Cox proportional hazards regression model was to assess the relative contribution of key prognostic genes to patient survival prediction. We visualized the univariate Cox regression analysis using the R-forestplot and R-survival packages and created a nomogram based on the optimal multivariate Cox regression analysis to predict 1-year, 3-year, and 5-year survival probabilities using the R-survival, R-rms, and R-ggplot2 packages. The concordance index (C-index) and calibration plot were used to evaluate the quality of the nomogram models. Using the R-survival package, decision curve analysis (DCA) was performed to evaluate the utility of the model and benefit to patients. The diagnostic value of hub genes was determined by calculating the ROC and the area under the curves (AUCs) [23]. Statistically, gene expression levels above or below the median value were designated as gene-high and gene-low, respectively.

### 2.6. Identification of HBx-Interacting Protein 

The human HCC cell line Huh-7 and the immortalized normal liver cell line HL-7702 (LO2) were obtained from the Chinese Academy of Sciences’ Institute of Biochemistry and Cell Biology (Shanghai, China). To generate stable HBx-overexpression cell lines (LO2-HBx, Huh-7-HBx), lentiviral vectors pLenti-HBx-HA-Zeo (Gene Chem, Shanghai, China) were constructed and used for corresponding cells by lentivirus-mediated transfection. G418 (Gene Chem, China) was used for 14 days to select stable transfections. The efficiency of HBx overexpression was validated using Western blotting (WB). LO2-HBx and Huh-7-HBx cells were harvested and lysed on ice for 30 min in RIPA buffer containing protease inhibitors. At 4 °C, the cell lysate was centrifuged at 12,000 rpm for 20 minutes. A small quantity of denatured supernatant was used for detection of HBx with WB. Next, 20 μL of protein A/G magnetic beads were added and incubated for 4 h. HBx antibody (22741, Genetex Inc, Irvine, CA, USA) was added to the cell lysate and incubated overnight at 4 °C. The immune precipitate was collected using centrifugation at 2000× *g* for 5 min. The supernatant containing HBx and its interacting proteins were separated using SDS-PAGE, and the separated products were digested and desalted prior to mass spectrometric identification. Liquid chromatography–tandem mass spectrometry (LC–MS/MS) analysis was conducted using OE Biotech (Shanghai, China). The Uniprot-Mus musculus database was used for mass spectrometric data analyses (MS1 tolerance: 10 ppm; MS2 tolerance: 0.05 Da; missed cleavage: 2) [24,25].

## 3. Results

Flowcharts were used to depict the overarching concept and methodology of this study (Figure 1).

### 3.1. Identification of DEGs Associated with HBV-Related HCC

Using PCA and UMAP, only two clusters were identified, and the accuracy of the groups’ characterization was confirmed by observing that most patients within the group were assigned to the same cluster (Figure 2A,B). In the GSE121248 dataset, GEO2R analysis revealed that 40 DEGs were up-regulated and 146 DEGs were down-regulated, while the R-limma method of analysis revealed 31 up-regulated DEGs and 91 down-regulated DEGs (Figure 2C,D). Figure 2E,F and Appendix A represent the candidate DEGs, including 89 down-regulated DEGs, which comprised 89 down-regulated genes and 31 up-regulated DEGs.

### 3.2. Functional Annotation of DEGs by GO and KEGG Pathway Analyses

GO analysis suggested that candidate DEGs were enriched in enzyme inhibitor activity, cyclin-dependent serine/threonine kinase regulator activity, cell cycle G1/S phase transition, liver development, response to anticancer agents, and hepatocyte differentiation (Figure 3A,B). As illustrated in Figure 3C,D, KEGG annotation analysis revealed that DEGs were significantly enriched in chemical carcinogenesis, the p53 signaling pathway, the PPRA signaling pathway, and cytochrome-p450-mediated xenobiotic metabolism.

### 3.3. Construction of PPI and Hub Gene Screening 

The PPI network was observed with 120 DEGs using the STRING database (Figure 4A). As illustrated in Figure 4B and Appendix A, 20 hub genes were screened at critical positions using the Cytohubba plug-in: *Anillin*, *Actin Binding Protein* (*ANLN*), *Cyclin B1* (*CCNB1*), *BUB1 Mitotic Checkpoint Serine/Threonine Kinase B (BUB1B)*, *Epithelial Cell Transforming 2* (*ECT2*), *Rac GTPase Activating Protein 1* (*RACGAP1*), *Estrogen Receptor 1* (*ESR1*), *PDZ Binding Kinase* (*PBK*), *DNA Topoisomerase II α* (*TOP2A*), *Cyclin Dependent Kinase 1* (*CDK1*), *Assembly Factor For Spindle Microtubules* (*ASPM*), *Ribonucleotide Reductase Regulatory Subunit M2* (*RRM2*), *NIMA Related Kinase 2* (*NEK2*), *Protein Regulator Of Cytokinesis 1* (*PRC1*), *Decorin* (*DCN*), *Insulin Like Growth Factor 1* (*IGF1*), *C-X-C Motif Chemokine Ligand 12* (*CXCL12*), *Hepatocyte Growth Factor* (*HGF*), *Secreted Phosphoprotein 1* (*SPP1*), *Hyaluronan Mediated Motility Receptor* (*HMMR*), and *Denticleless E3 Ubiquitin Protein Ligase Homolog* (*DTL*). 

### 3.4. Validation of Expression Patterns of Hub Genes 

In total, 15 hub genes (*CCNB1*, *CDK1*, *BUB1B*, *ECT2*, *RACGAP1*, *ANLN*, *PBK*, *TOP2A*, *ASPM*, *RRM2*, *NEK2*, *PRC1*, *SPP1*, *HMMR*, and *DTL*) were significantly up-regulated in tumors relative to adjacent normal tissues in TCGA database (Figure 5A,B). The analysis of the co-expression models for hub genes revealed that 15 up-regulated hub genes had a strong positive relationship with one another (Figure 5C). The strongest correlation was observed for TOP2A and HMMR, as well as for ANLN and CDK1 and for BUB1B and RACGAP1, with a correlation coefficient of 0.94. The KEGG pathways results for the 15 up-regulated hub genes revealed that these genes were significantly enriched in the same signaling pathways, particularly in the p53 signaling pathway (Figure 5D). Moreover, these 15 hub genes depicted an upward trend in the progression from liver cirrhosis to HCC in the GEO database (Figure 6).

### 3.5. Optimal Diagnostic and Predictive Biosignature of Hub Genes for HBV-Related HCC

Patients with high expression of 15 hub genes (*CCNB1*, *CDK1*, *BUB1B*, *ECT2*, *RACGAP1*, *ANLN*, *PBK*, *TOP2A*, *ASPM*, *RRM2*, *NEK2*, *PRC1*, *SPP1*, *HMMR*, and *DTL*) were found to have a poorer OS, indicating these genes are important prognostic markers (Figure 7A). The remaining five genes, however, exhibited no significant differences. The risk score model revealed that as the risk value increases, patient survival time tends to decrease, whereas the fatality rate tends to rise, and the genes tend to be highly expressed (Figure 7B). The forest plot of univariate Cox regression is depicted in Figure 7C. As shown in Figure 7D, we developed a nomogram to estimate and manage patient efficacy based on the probability of individual survival.

The C-index of 0.702 of the nomogram indicated that the prediction and observation results matched perfectly [26,27,28]. The calibration plot revealed a straight line approaching the diagonal, indicating a strong correlation between the actual and estimated probabilities at 1.3 and 5 years (Figure 8A). DCA revealed that the nomogram model had a greater net benefit and a broader threshold probability than other independent predictors, indicating that it had the most clinical utility (Figure 8B). As shown in Figure 8C, the AUCs of *CCNB*, *CDK1*, *BUB1B*, *ECT2*, *RACGAP1*, *ANLN*, *PBK*, *TOP2A*, *ASPM*, *RRM2*, *NEK2*, *PRC1*, *SPP1*, *HMMR*, and *DTL* were, respectively, 0.980, 0.978, 0.962, 0.946, 0.980, 0.967, 0.964, 0.973, 0.972, 0.961, 0.971, 0.981, 0.724, 0.969, and 0.961, which indicated that these 15 hub genes could efficiently discriminate tumor and normal samples and show a satisfactory performance in survival prediction. 

### 3.6. Identification of a Set of Novel HBx-Interacting Proteins

WB results indicated that HBx was present in the immunoprecipitated mixture of LO2-HBx and Huh-7-HBx cells, respectively, which demonstrated the efficiency of the Co-IP strategy (Appendix A). Coomassie-stained gels revealed that several bands appeared only in the LO2-HBx lane and the Huh-7-HBx lane but not in the control vector lanes (LO2-HBx IgG and Huh-7-HBx IgG; Figure 9A). Mass spectrometry (MS) was performed on the HBx-interacting protein complex extracted from LO2-HBx cells and Huh-7-HBx cells (Appendix A). Co-IP/MS experiments uncovered 2531 HBx-interacting proteins in LO2-HBx cells and 2393 proteins in Huh-7-HBx cells (Figure 9B and Table 1). CDK1, RRM2, ANLN, and HMMR were the proteins in the two HBx-overexpressing cell models, and they were identified as candidate HBx downstream proteins (Figure 9C,D).

## 4. Discussion

The prevalence of clinical genetic testing has led to an accurate description of the relationship between genes and diseases, which enables the implementation of genetic testing in healthcare on an unprecedented scale, both for early disease detection and for disease prevention. Advances in genomics and proteomics for the detection of genetic biomarkers have demonstrated great promise for the early diagnosis of HCC [29,30]. In this study, we analyzed the GSE121248 dataset using two methods to identify the DEGs involved in the progression of HBV infection to HCC. Prior to this, we used the clustering techniques PCA and UAMP to ensure that there were no clusters with similar intrinsic characteristics between the HBV-infected HCC tissue group and the para-cancer liver tissue group, which could improve the credibility of the grouping [31]. A set of 120 candidate DEGs (31 up-regulated DEGs and 89 down-regulated DEGs) was identified. Enrichment analysis of the candidate DEGs revealed that they are robust during the transformation of HBV-infected hepatocytes to HCC on the cellular metabolic pathway and higher biological functions [32]. Proteins frequently perform their functions by forming complexes. HBx induces promiscuous transactivation by means of protein-protein interaction [33]. To gain a better understanding of how these distinct molecules contribute to the prevalence of HBV-related HCC, the physical and functional interactions between these proteins were analyzed and 20 pivotal molecules (hub genes) that contribute significantly to the interaction were screened using the STRING database. Using TCGA HCC datasets, the expression levels of the 20 hub genes were reconfirmed. Co-expression model analysis revealed strong positive correlations between 15 up-regulated hub genes (*CCNB1*, *CDK1*, *BUB1B*, *ECT2*, *RACGAP1*, *ANLN*, *PBK*, *TOP2A*, *ASPM*, *RRM2*, *NEK2*, *PRC1*, *SPP1*, *HMMR*, and *DTL*), suggesting their involvement in the same signaling pathway governing the occurrence of HBV-related HCC. This is consistent with the results of the GO and KEGG analyses, which showed that integrated DEGs are especially abundant in the p53 signaling cascade. Previous research has theorized that the p53 signaling pathway modulates hepatocarcinogenesis by regulating HCC inhibition, apoptosis, senescence, and DNA damage [34]. Meanwhile, in the progression from cirrhosis to HCC, these 15 hub genes also demonstrated a tendency toward elevated expression. Consequently, these 15 hub genes are likely involved in the progression of the disease from HBV infection to HCC, and exploring these hub genes will aid in predicting tumorigenesis. On the basis of the median level of hub gene expression in HCC, the hub genes were classified as either high expression or low expression. We plotted prognostic Kaplan-Meier survival curves for the 20 hub genes to assess their clinical utility. The findings demonstrated that hub genes in the high-expression group could predict poor OS. The risk score model confirmed that these 15 hub genes were typically highly expressed in patients with an elevated HCC risk, indicating that they may be HCC risk factors. The nomogram could accurately predict the prognosis of patients with HCC and assist clinicians in evaluating the risk associated with each patient with HCC. A combination of the ROC curve, C-index, and DCA confirmed the predictive value of the nomogram. In addition, ROC analysis confirmed the predictive classification effect of the HCC risk score model and the diagnostic significance of hub genes for HCC.

Co-IP/MS analysis of HBx-interacting proteins was performed, for the first time, to validate the key molecules involved in the carcinogenic progression of HCC disease. We discovered that four key genes (*CDK1*, *RRM2*, *ANLN*, and *HMMR*) in the prediction model interacted with HBx during the entire disease process, including HBV infection and HCC occurrence. Previous research had established associations between these four potential key molecules and the occurrence of HCC, which our results confirmed. For instance, HMMR, one of a handful of known hyaluronan receptors, may play a crucial role in the neoplastic progression of diverse tumors and thus may be considered a multifunctional oncogenic protein [35,36]. According to a recent review, HMMR is low in the majority of healthy tissues but increases in hyperplastic tissues. HMMR is overexpressed in HCC tissues relative to normal tissues, and its level is associated with a poor prognosis [37]. CDK1 is a well-characterized checkpoint protein involved in cell proliferation and transcriptional control [38]. Previous research has established a link between CDK1-mediated deregulation of cell proliferation and the development of HCC [39]. Additionally, the CDK1/cyclin B1 (CCNB1) axis can be regulated to influence the development of HCC [40]. It was reported that CDK1 overexpression is related with the clinical and pathological characteristics of patients with HCC and a bad prognosis. Down-regulation of CDK1 during the G2/M phase transition in HCC cells inhibits cell cycle progression. It has been theorized that overexpression of CDK1 in HCC is associated with cell cycle abnormality and that inhibiting CDK1 expression increases hepatoma cell senescence and autophagy [41]. RRM2 is required for tumor growth and functions as a tumor drive [42]. It was established that RRM2 is a powerful inhibitor of HCC cell proliferation and HBV replication [43]. According to a previous study [44], RRM2 promotes the synthesis of GSH in HCC cells, acting as an anti-ferroptotic factor. Serum RRM2 is valuable as a biomarker for assessing the extent to which ferroptosis is suppressed, thereby improving the diagnostic efficiency of liver cancer. ANLN is an actin-binding protein necessary for cleavage furrow formation during cytokinesis [45]. ANLN is essential for cell division, differentiation, adhesion, migration, apoptosis, and cycle progression [46]. It has been reported that the CDK1-PLK1-SGOL2-ANLN signaling axis, which mediates aberrant cell cycle division, may play a critical role in the development of HCC [47]. The silencing of ANLN may inhibit HCC cell proliferation in the G2/M phase, as well as their migration and invasion capacities. In contrast, the expression of cytosolic ANLN was significantly higher in the metastatic foci of HCC than in the primary tumor tissue. Moreover, cytosolic ANLN expression was an independent risk factor for the 5-year OS following hepatectomy [48].

A suite of biomarkers with high diagnostic sensitivity and specificity can be used as for ancillary detection in current HCC-screening methods in primary care to ameliorate the current situation of a low HCC detection rate, particularly for AFP-negative patients with small nodules, whose diagnosis is challenging. The combination of liver biopsy and biomarker detection results by IHC or PCR to assess the disease status of patients could improve the disease detection rate of patients and formulate effective follow-up and intervention strategies. We discovered 15 hub genes that are specifically expressed in HBV-related HCC and confirmed a pattern of increased expression of these hub genes as the disease progresses from cirrhosis to HCC. Meanwhile, based on TCGA database, a preliminary clinical value evaluation of these important genes for hepatocellular carcinoma was carried out. Our findings offer new chemoprevention and diagnostic approaches for HCC surveillance in patients with cirrhosis receiving primary care, and they also point the way for future study. This will serve to further increase the accuracy of primary HCC surveillance and patients’ confidence so as to further realize the popularization of liver cancer screening in primary care and start a virtuous circle. 

Except for liver cancer, many tumors have no obvious clinical symptoms in the early and middle stages of development, and their development is characterized by occult characteristics, such as cancer of the pancreas [49]. By the time the patients present with symptoms, such as pain and jaundice, they are already in the middle and advanced stages of the disease, and the natural course of this disease is only 3 to 6 months. Although this study was limited to HBV-related HCC, it could serve as an example for other diseases, such as cancers we encounter in PCP practice, and may provide research directions and insight for the early detection of occult tumors in primary medical surveillance.

Inevitably, some limitations remain in this study. For instance, before the key hub genes we have identified can be applied as diagnostic biomarkers for hepatocellular carcinoma, larger sample sizes from various clinical centers should be used to confirm their expression trends and further investigation of their functions and mechanisms involved in hepatocarcinogenesis is needed.

## 5. Conclusions

Personalized medicine will ultimately be based on a collection of diagnostic and prognostic biomarkers. Our team identified potential diagnostic biomarkers involved in the progression of cirrhosis to HCC from a carcinogenic perspective and built a disease prediction model for HBV. The findings may inspire new ways for screening and monitoring patients with a high risk of cirrhosis, as well as those who currently have HCC.

## Figures and Tables

**Figure 1 genes-13-02331-f001:**
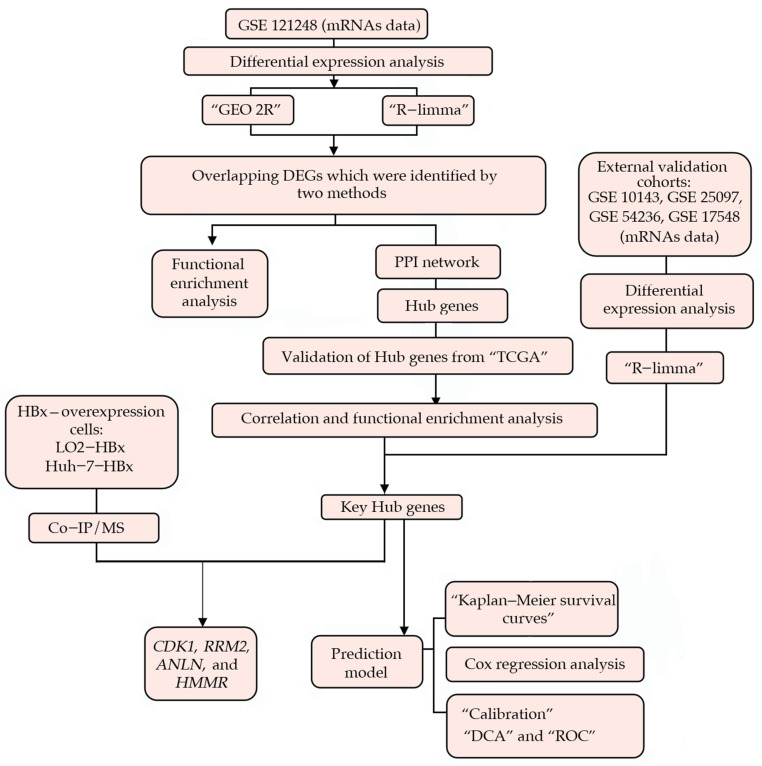
The workflow diagram of data acquisition, preprocessing, analysis, and validation. Two distinct techniques were used to identify DEGs in the GSE121248 dataset. Among the DEGs, a PPI network was constructed, and Cytoscape was used to extract hub genes. The TCGA database validated the expression levels, correlation analysis, and predictive performance of hub genes. Using the GSE10143, GSE25097, GSE54236, and GSE17548 datasets, the expression pattern of hub genes during the progression from cirrhosis to HCC was analyzed. A prediction model was then developed using Cox regression analysis. In both cell models, CDK1, RRM2, ANLN, and HMMR interacted specifically with HBx, as determined by Co-IP/MS. DEGs, differentially expressed genes; Co-IP/MS, co-immunoprecipitation coupled with mass spectrometry; PPI, protein–protein interaction.

**Figure 2 genes-13-02331-f002:**
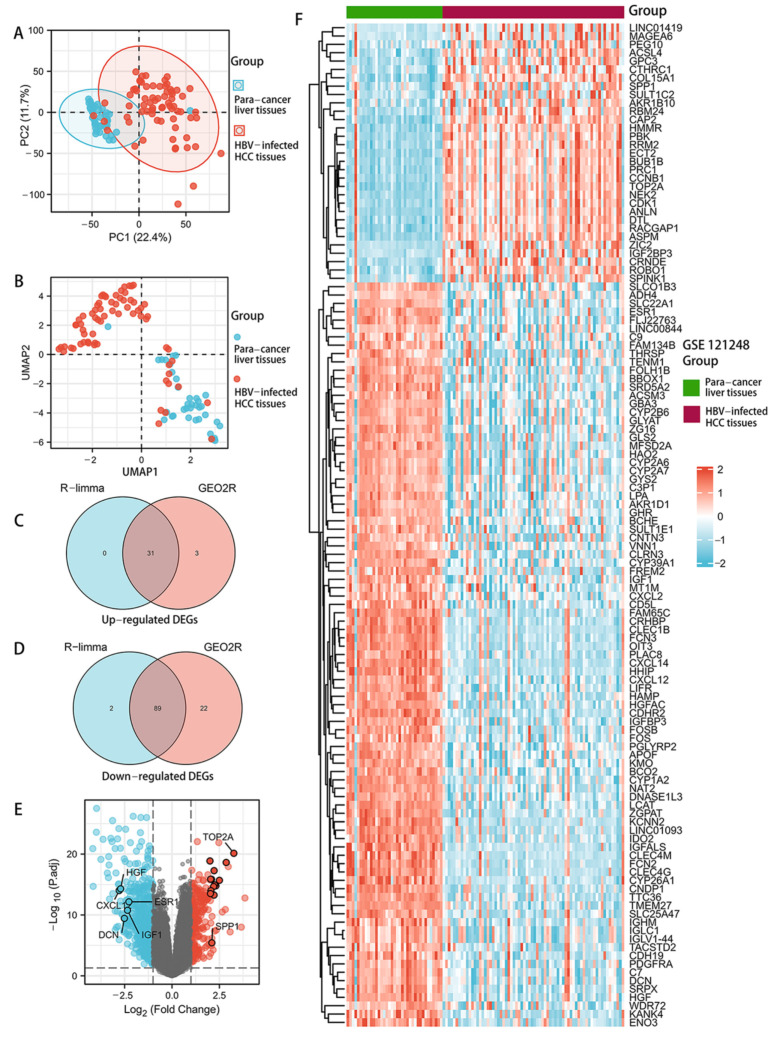
Exploration of potential DEGs in the GSE121248 dataset. The PCA plot (**A**) and UMAP clustering plot (**B**) based on samples from the para-cancer liver tissue group and the HBV-infected HCC tissue group. The distance of sample point clusters indicates that they come from different groups. The Venn diagram of up-regulated DEGs (**C**) and down-regulated DEGs (**D**) between the two methods. Volcano plots of DEGs (**E**). Blue and red indicate down-regulated overlapping DEGs and up-regulated overlapping DEGs, respectively. The expression profiles of overlapping DEGs between HBV-infected HCC tissues and para-cancer liver tissues were visualized in a heatmap (**F**).

**Figure 3 genes-13-02331-f003:**
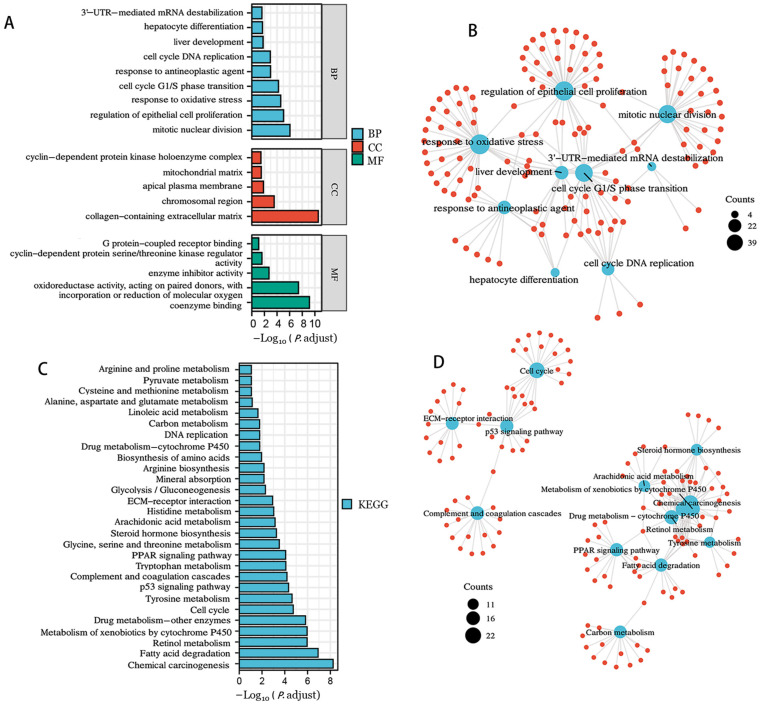
GO and KEGG enrichment analyses of differentially expressed genes in HBV-related HCC. GO term enrichment results (**A**,**B**). KEGG enrichment results (**C**,**D**). The blue nodes represent the entries, the red nodes represent the numerators, and the connecting lines represent their relationship.

**Figure 4 genes-13-02331-f004:**
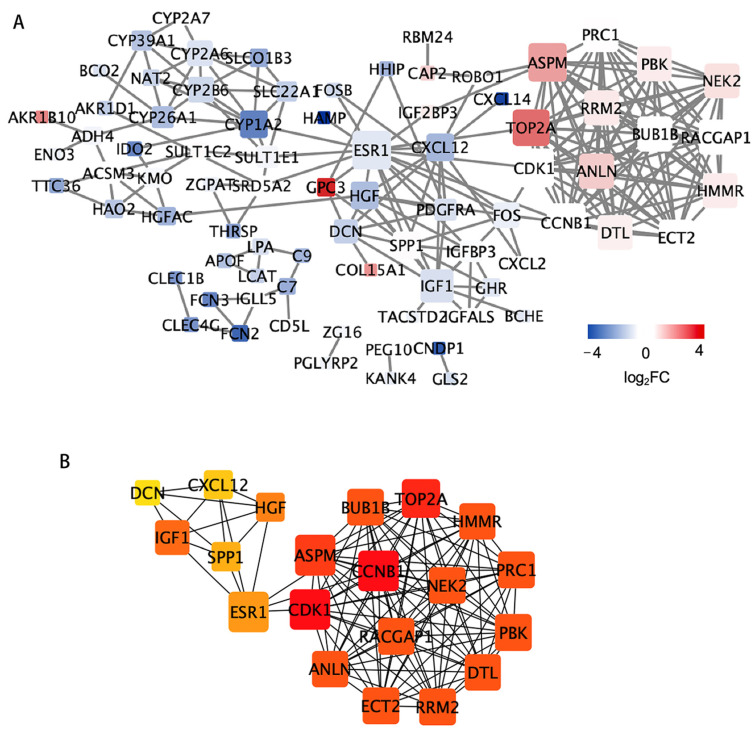
PPI network and module analysis. PPI network construction for candidate genes using STRING11.0 was visualized using Cytoscape (**A**). The size of the node refers to the degree standard, and the color of the node represents the values of log_2_FC. The network directly associated with the top 20 hub genes identified using CytoHubba (**B**). The color depth of the nodes represents their importance in the network.

**Figure 5 genes-13-02331-f005:**
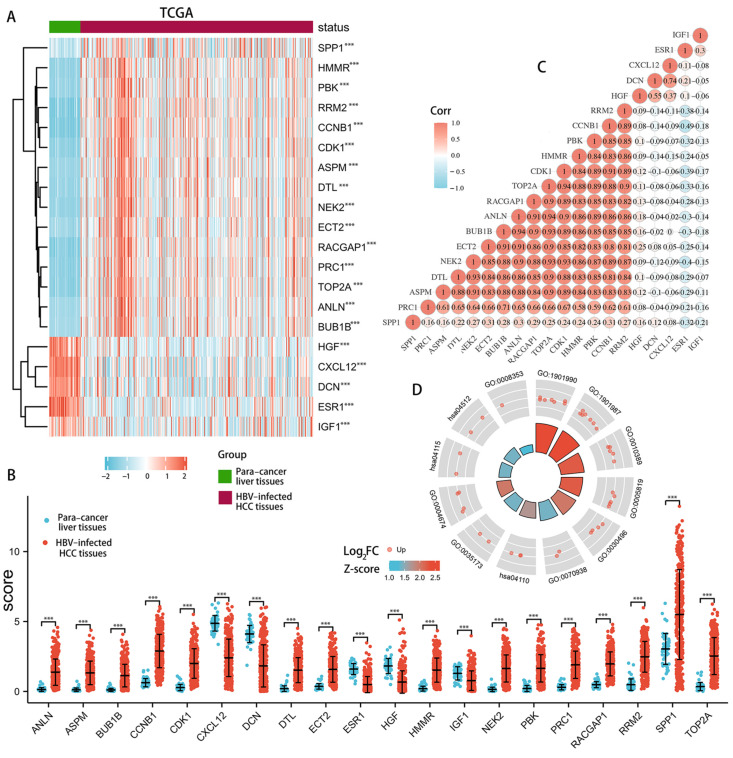
Expression validation of candidate hub genes in TCGA database. Heatmaps of hub genes expressed in tumors and adjacent normal tissues in patients with HCC (**A**). Validation of the expression level of hub genes in patients with HCC (**B**). The correlation matrix of interaction in hub genes. Correlation coefficients are plotted with negative correlation (blue) and positive correlation (red) (**C**). Functional enrichment analysis of the up-regulated hub genes (**D**). GO 1901990: regulation of mitotic cell cycle phase transition; GO 1901987: regulation of cell cycle phase transition; GO 0010389: regulation of G2/M transition of the mitotic cell cycle; GO 0005819: spindle; GO 0030496: midbody; GO 0070938: contractile ring; GO 0004674: protein serine/threonine kinase activity; GO 0008353: RNA polymerase II CTD heptapeptide repeat kinase activity; hsa04110: cell cycle; hsa04115: p53 signaling pathway; hsa04512: ECM–receptor interaction. *** *p* < 0.001.

**Figure 6 genes-13-02331-f006:**
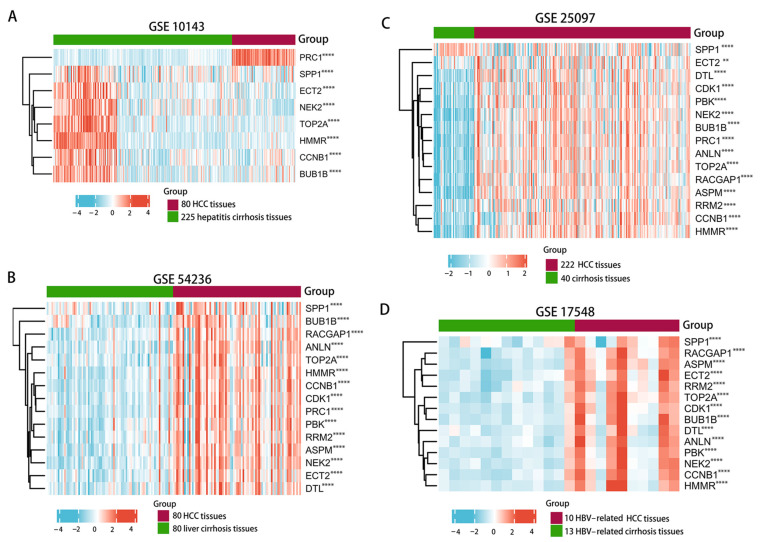
Expression validation of 15 up-regulated hub genes between liver cirrhosis and HCC tissues in the GEO database. HCC tissue group vs. cirrhosis tissue group in GSE10143 (**A**), HCC tissue group vs. cirrhosis tissue group in GSE54236 (**B**), HCC tissue group vs. cirrhosis tissue group in GSE25097 (**C**), and HBV-related HCC tissue group vs. HBV-related cirrhosis tissue group in GSE17548 (**D**). Green and red represent cirrhosis and HCC samples, respectively. GEO, Gene Expression Omnibus. ** *p* <0.01, **** *p* < 0.0001.

**Figure 7 genes-13-02331-f007:**
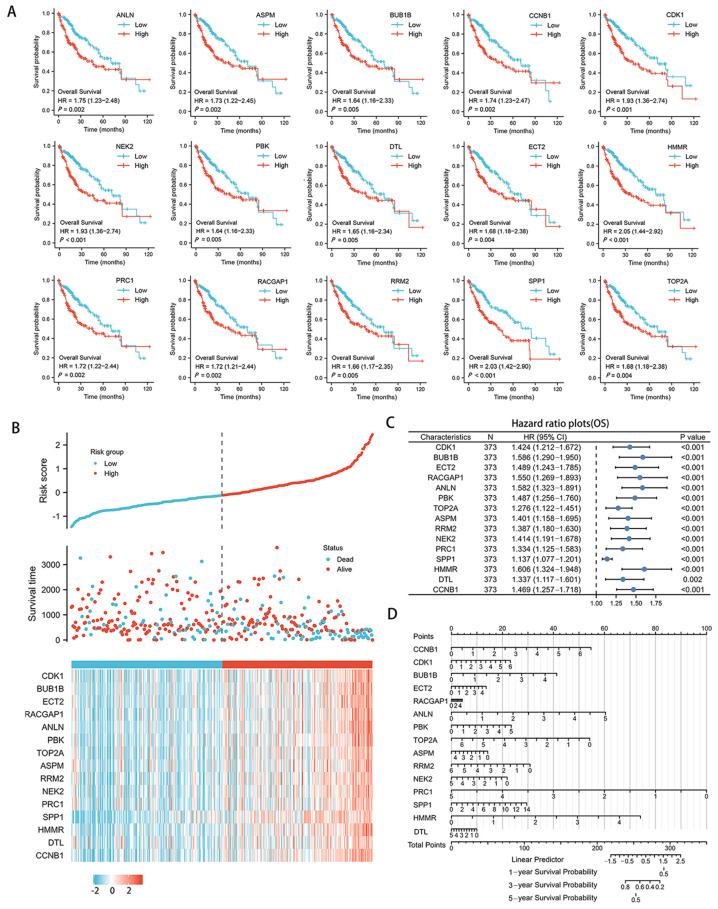
Multifaceted clinical value analysis of up-regulated hub genes in HCC. (**A**) Kaplan-Meier plot of OS between high-expression and low-expression groups. The red lines and blue lines, respectively, represent patients with high gene and low gene expression. (**B**) Risk score in the cohort of TCGA. The distribution of patients with an increased risk score into high- and low-risk groups, as well as scatter plots depicting patient survival with an increased risk score. (**C**) Forest plot showing OS after univariate Cox regression analysis in HCC. (**D**) Nomogram for predicting the probability of 1-, 3-, and 5-year OS for patients with HCC. OS, overall survival.

**Figure 8 genes-13-02331-f008:**
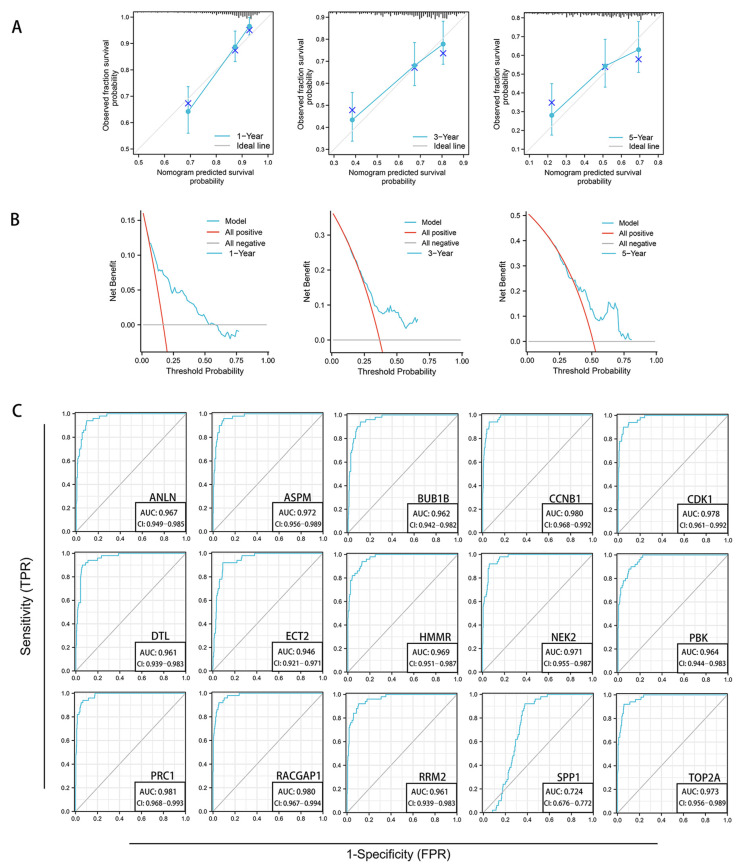
Assessment of precision, sensitivity, and clinical utility of the predictive model. (**A**) Calibration curve for predicting the probability of 1-, 3-, and 5-year OS for patients with HCC. Calibration curve of the nomogram in HCC from TCGA data. (**B**) DCA plots of the nomogram by calculating the C-index. (**C**) ROC curve analysis of 15 up-regulated hub genes to distinguish HBV-related HCC tissues from normal tissues. ROC, receiver operating characteristic; DCA, decision curve analysis.

**Figure 9 genes-13-02331-f009:**
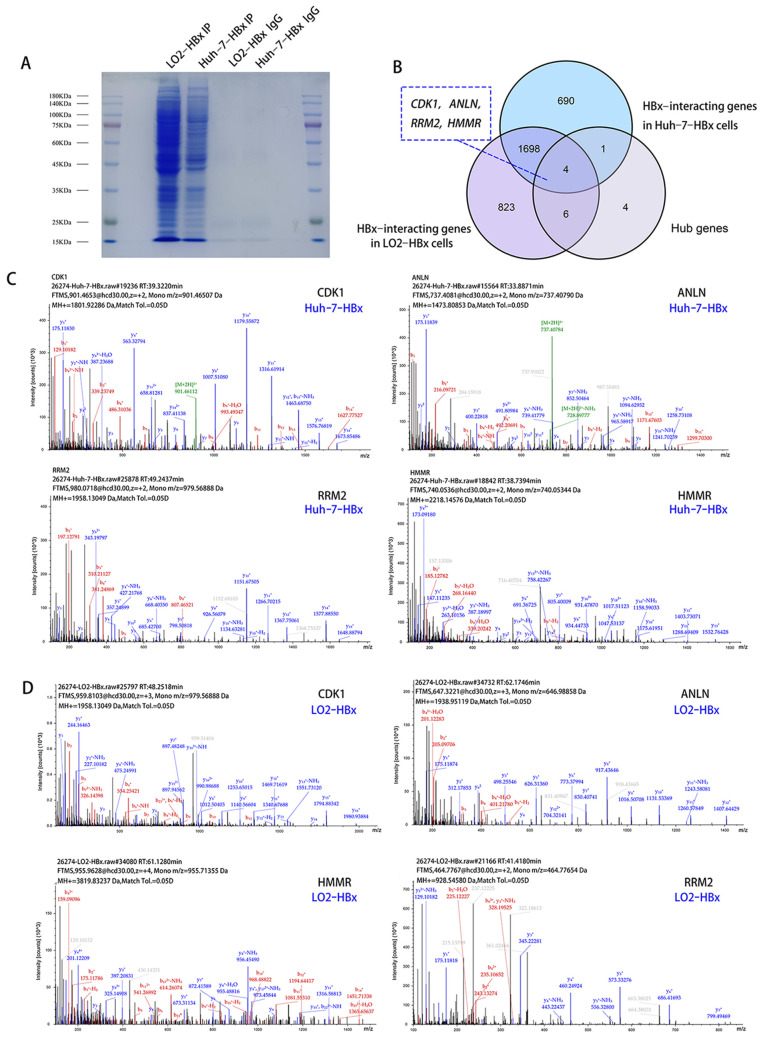
Identification of HBx-interacting proteins. (**A**) The co-immunoprecipitated mixture was separated using SDS-PAGE and stained with Coomassie blue. (**B**) Venn diagram of overlapping HBx-interacting proteins and hub genes. Identification of CDK1, RRM2, ANLN, and HMMR from the HBx-interacting protein complex extracted from LO2-HBx cells (**C**) and Huh-7-HBx cells (**D**) using Co-IP/MS analysis.

**Table 1 genes-13-02331-t001:** The list of identified HBx-interacting proteins from CO-IP/MS/MS.

Sample	Hub Genes	Score Sequest HT	Peptides	PSMs	Unique Peptides	Accession	Coverage (%)
LO2-HBx	*TOP2A*	23.51	8	8	4	P11388	6
	*RACGAP1*	18.17	5	5	5	Q9H0H5	16
	*CCNB1*	14.38	4	4	4	P14635	17
	*PBK*	10.73	3	3	3	Q96KB5	12
	*BUB1B*	9.35	3	3	3	O60566	5
	*PRC1*	4.6	1	1	1	O43663	4
	*CDK1*	117.95	18	32	16	P06493	70
	*RRM2*	1.87	1	1	1	P31350	2
	*ANLN*	7.51	2	2	2	Q9NQW6	3
	*HMMR*	17.13	2	3	2	O75330	8
Huh-7-HBx	*ECT2*	2.67	1	1	1	Q9H8V3	1
	*CDK1*	73.97	15	21	13	P06493	63
	*RRM2*	6.8	2	2	2	P31350	8
	*ANLN*	6.51	2	2	2	Q9NQW6	3
	*HMMR*	3.86	1	1	1	O75330	3

## Data Availability

All the datasets used in this research are accessible via online public databases.

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
