# Peer review of "Significance of Identifying Key Genes Involved in HBV-Related Hepatocellular Carcinoma for Primary Care Surveillance of Patients with Cirrhosis"

_genes, 2022, doi:10.3390/genes13122331_

Round 1

Reviewer 1 Report

- References needed o be updated where it include only two references in 2022

- The manuscript need to be reviewed by an English native speaker where it include minutes errors .

Author Response

Point 1(Reviewer 1)

References needed to be updated where it include only two references in 2022

Response 1:

Thank you very much for your comments. Following a rigorous literature search and analysis, we have updated the citations.

Point 2(Reviewer 1)

- The manuscript need to be reviewed by an English native speaker where it include minutes errors .

Response 2:

We invited the native English speaker to help language polishing, and hope our improvements of language could meet the reviewers’expectation.

Reviewer 2 Report

Dear Editor of Genes,

Sorry for my late submission and thanks so much for extending my reviewing time.

The authors of this manuscript raise a high aim to identify the key genes of HBV-related Hepatocellular Carcinoma. However, their data only demonstrated the differential gene expressions in distinct HCC and cirrhosis datasets.  On the basis of the data they show, the authors do find some differential expression genes and hubs in the PPI network. However, the evidence they show in the manuscript, including the differential expression in the distinct dataset and the involvement of hub proteins during the transformation of HCC, is not valid support of the reliability that these genes could predict HBV-related Hepatocellular Carcinoma. Unless the author could provide further solid support for their conclusion, I do not think the paper should be accepted at the current stage.

Best~

Feng

Author Response

Point 1(Reviewer 2): The authors of this manuscript raise a high aim to identify the key genes of HBV-related Hepatocellular Carcinoma. However, their data only demonstrated the differential gene expressions in distinct HCC and cirrhosis datasets.  On the basis of the data they show, the authors do find some differential expression genes and hubs in the PPI network. However, the evidence they show in the manuscript, including the differential expression in the distinct dataset and the involvement of hub proteins during the transformation of HCC, is not valid support of the reliability that these genes could predict HBV-related Hepatocellular Carcinoma. Unless the author could provide further solid support for their conclusion, I do not think the paper should be accepted at the current stage.

Response 1:

Thank you very much for your comments.

Using experimental data from the five following areas, we demonstrate that the 15 increased hub genes we identified have early warning implications for hepatocarcinogenesis:

1)-As depicted in Figures 5A and 5B, 15 Hub genes were highly upregulated in hepatocellular carcinoma tissues relative to paracancerous tissue, demonstrating that Hub genes possess oncogene capabilities. Our team had validated the oncogenic role of RRM2 in HCC, indicating the validity of our research and the credibility of our findings. And "Peer J" has accepted this article " A pan-cancer analysis of the oncogenic role of ribonucleotide reductase subunit M2 in human tumors " in which the results of the study are reported.

Figure 5. Expression validation of candidate Hub genes in TCGA database. Heatmaps of Hub genes expressed in tumors and adjacent normal tissues in HCC patients (A). Validation of the expression level of Hub genes in HCC patients (B). The correlation matrix of interaction in Hub genes. Correlation coefficients are plotted with negative correlation (blue) and positive correlation (red) (C). Functional enrichment analysis of the up-regulated Hub genes (D).

2)-One of the current strategies for the clinical diagnosis of hepatocellular carcinoma is measuring the AFP expression levels of patients. AFP test findings can assist clinicians in verifying a patient's diagnosis of liver cancer when elevated AFP levels indicate a high risk for liver cancer.

Therefore, we investigated the expression of 15 Hub genes in cirrhotic and hepatocellular carcinoma samples and discovered that Hub genes were expressed at higher levels in HCC than in cirrhosis, suggesting the early warning function of Hub genes for HCC (Figure 6).

Figure 6. Expression validation of 15 up-regulated Hub genes between liver cirrhosis and HCC tissues in GEO databases. HCC tissues group vs. cirrhosis tissues group in GSE10143 (A), HCC tissues group vs. cirrhosis tissues group in GSE54236 (B), HCC tissues group vs. cirrhosis tissues group in GSE25097 (C), HBV-related HCC tissues vs. HBV-related cirrhosis tissues group in GSE17548 (D). Green and red represents cirrhosis and HCC samples, respectively. GEO, Gene Expression Omnibus.

3)- Receiver operating characteristic (ROC) curves are frequently employed to evaluate the validity of a metric for identifying or diagnosing two groups of test participants (e.g., patients and normal individuals). In Figure 8C, we assessed the diagnostic value of each molecule for hepatocellular carcinoma using ROC analysis for each molecule. All 14 Hub genes, with the exception of SPP1, exhibited AUC values greater than 0.90, indicating a high diagnostic value for HCC.

Figure 8. Assessment of precision, sensitivity, and clinical utility of the predictive model. (A) Calibration curve for predicting the probability of 1-, 3-, and 5-year OS for HCC patients. And the calibration curve of the nomogram in HCC from TCGA data. (B) DCA plots of the nomogram by calculating the C-index. (C) ROC curve analysis of 15 up-regulated Hub genes to distinguish HBV-related HCC tissues from normal tissues. ROC, receiver operating characteristic; DCA, Decision curve analysis.

4)-As depicted in Figure 7, we evaluated the clinical significance of these 15 elevated Hub genes. K-M analysis revealed that all Hub genes corresponded to HR values greater than 1.5, and SPP1, TOP2, and HMMR corresponded to HR values greater than 2, indicating that Hub genes had a significant effect on survival time of patients and were a risk factor for survival of patients with HCC. Risk prediction models based on 15 highly expressed Hub genes showed that expression of each of these genes increased with increasing risk scores, implying that Hub gene expression levels play a key role in determining the severity and duration of a patient's condition.In addition, the analytical plots of the forest plot and nomogram indicate similarly the effect of the Hub gene on patient survival.

Figure 7. Multifaceted clinical value analysis of up-regulated Hub genes in HCC (A) Kaplan-Meier plot of OS between high-expression and low-expression groups. The red lines and blue lines respectively represent patients with high gene- and low gene- expression. (B) Risk score in the TCGA cohort. The distribution of patients with an increased risk score into high and low risk groups, as well as scatter plots depicting patient survival with an increased risk score. (C) Forest plot showing OS after univariate Cox regression analysis in HCC. (D) Nomogram for predicting the probability of 1-, 3-, and 5-year OS for HCC patients. OS, overall survival.

Reviewer 3 Report

Authors identified the potential key genes that are involved in HBV related HCC and also demonstrated using proteomic analysis, these key Hub genes interact with the HBV-encoded oncogene X protein (HBx), the oncogenic protein in HCC.

Overall, the manuscript is well written and well structured.

Though there are several papers regarding kye genes that are involved in HBV related HCC (PMID: 31392101, PMID: 32269621, PMID: 30864827).

In this manuscript authors with the help of proteomic analysis showed the interaction of these hub genes with HBX which is the intersecting part.

In my opinion it will be great if authors will identify the downstream signaling pathways. Also using human HBV related HCC samples, they can analyze the status of those key hub genes.

Author Response

Point 1(Reviewer 3): Authors identified the potential key genes that are involved in HBV-related HCC and also demonstrated using proteomic analysis, these key Hub genes interact with the HBV-encoded oncogene X protein (HBx), the oncogenic protein in HCC. Overall, the manuscript is well written and well structured. Though there are several papers regarding key genes that are involved in HBV related HCC (PMID: 31392101, PMID: 32269621, PMID: 30864827).

In this manuscript authors with the help of proteomic analysis showed the interaction of these hub genes with HBX which is the intersecting part.

In my opinion it will be great if authors will identify the downstream signaling pathways. Also using human HBV related HCC samples, they can analyze the status of those key hub genes. 观点1

Response 1:

Thank you very much for your comments and constructive suggestions.

Our team identified potential Hub genes involved in HBV-associated HCC and also used proteomic analysis to demonstrate for the first time the role of these Hub genes in hepatocarcinogenesis at the level of oncogenicity.

Prior to identifying downstream signaling pathways, we conducted a correlation study of the Hub gene.  Analysis of co-expression model revealed a substantial positive correlation between the 15 elevated crucial genes, indicating that they interact or may be involved in the same oncogenic signaling pathway (Figure 5C). On this basis, we did GO and KEGG analyses on these 15 Hub genes, and the findings indicated that the signaling pathways heavily enriched by Hub genes were cell cycle, p53 signaling pathway, and ECM-receptor interaction (Figure 5D).

Figure 5. Expression validation of candidate Hub genes in TCGA database. Heatmaps of Hub genes expressed in tumors and adjacent normal tissues in HCC patients (A). Validation of the expression level of Hub genes in HCC patients (B). The correlation matrix of interaction in Hub genes. Correlation coefficients are plotted with negative correlation (blue) and positive correlation (red) (C). Functional enrichment analysis of the up-regulated Hub genes (D). GO 1901990: regulation of mitotic cell cycle phase transition, GO 1901987: regulation of cell cycle phase transition, GO 0010389: regulation of G2/M transition of mitotic cell cycle, GO 0005819: spindle, GO 0030496: midbody, GO 0070938: contractile ring, GO 0004674: protein serine/threonine kinase activity, GO 0008353: RNA polymerase II CTD heptapeptide repeat kinase activity, hsa04110: Cell cycle, hsa04115: p53 signaling pathway, hsa04512: ECM-receptor interaction.

As for the expression of these Hub genes in HBV-related HCC, we can find from the analysis results in Figure 6D that SPP1, RACGAP1, ASPM, ECT2, RRM2, TOP2A, CDK1, BUB1B, DTL, ANLN, PBK, NEK2 in the GSE17548 dataset, CCNB1, and HMMR were less expressed in HBV-related cirrhotic samples than in HBV-associated hepatocellular carcinoma samples.

For PRC1, there are currently no human HBV-related hepatocellular carcinoma sample datasets demonstrating its expression. However, hepatocellular carcinoma datasets are available for analyzing its expression. In the HCC datasets GSE54236, GSE10143, and GSE25097, the expression of PRC1 was statistically higher in hepatocellular carcinoma tissues than in cirrhosis tissues (Figure 6A-C).

Figure 6. Expression validation of 15 up-regulated Hub genes between liver cirrhosis and HCC tissues in GEO databases. HCC tissues group vs. cirrhosis tissues group in GSE10143 (A), HCC tissues group vs. cirrhosis tissues group in GSE54236 (B), HCC tissues group vs. cirrhosis tissues group in GSE25097 (C), HBV-related HCC tissues vs. HBV-related cirrhosis tissues group in GSE17548 (D). Green and red represents cirrhosis and HCC samples, respectively. GEO, Gene Expression Omnibus.

Round 2

Reviewer 2 Report

The authors of this paper identified 120 differential expressed genes in HBV-related Hepatocellular Carcinoma dataset (GSE121248). Further, A protein-protein interaction (PPI) network was build to identified hubs in the network. The authors declared that these hub genes would be used to build a prediction model. Further, proteomics data demonstrated that 4 hub genes are involved in the transformation process of HCC after HBV infection. However, the authors fail to provide some important details of their data analysis and an overall evaluation of their model, which is very important if clinicans would like to use their model to help the corresponding diagnosis. For that reason, I would suggest that the following is undertaken to strengthen the existing model and support the conclusions made, before considering the manuscript for potential publication:

Major:

1.       The aim of this project is identify the key genes involveld in HBV-Related Hepatocellular Carcinoma. In the manuscript, the key genes are selected and validated on the basis of the comparision between HCC and cirrhosis. The author need to explain the rational why they focus on the comparison between HCC and cirrhosis since the HCC is a complicated disease and so many distinct factors could trigger the pathogenesis of HCC in so many differenct perspective;

2.       The details of prediction model is very important to support the authors’ aim of providing more precision diagnosis of HCC on the gene levels. The details of the prediction model should be added to the revised version of this paper;

3.       The details of using the Nomogram to predict the overall survival is missing in this manuscript. I did see the authors used the package “R-forestplot” to creat a nomogram. Since Nomogram provide very important support that these hub genes could predict the survival of distinct patients, the details of buidling and using Nomogram should be further provided at least in the supplementary methods;

4.       Traditionaly, C-index of Nomogram larger than 0.85 is considered the reliable prediction of survival probability. If the author insist that 0.702, which is lower than 0.85, is already good enough, the authors should demonstrate the corresponding evidences;

5.       The 120 DEGs and the hub genes are very important since this is the key discovery of this project. The authors should provide the list of these DEGs and hub genes in their manuscript;

6.       The authors mentioned that their genomics and proteomics data demonstrated great promise for the early diagnosis of HCC. If no direct evidence of better earlier diagnosis of HCC would be shown in the manuscript, I am afraid that the author should not make this narrative in their manuscript.

Minor:

1.       Both PCA and UMAP is used in this project, since not all the readers of Genes are statistician, the authors should demonstrate their rational of using both methods;

2.       In figure 3, the author shows the results of KEGG in their panel C and D, they should add more details such as what is the differences between the blue dots and red dots in their Figure Legend;

3.       This manuscript have some typos, such as in line 347, “DGEs” should be “DEGs”, further proofreading is needed;

4. The English of this manuscript still has space to improve. If possible, the authors should find a native speaker to proofread the whole manuscript.

Author Response

Response to Reviewer 2 Comments

Point 1: The authors of this paper identified 120 differential expressed genes in HBV-related Hepatocellular Carcinoma dataset (GSE121248). Further, A protein-protein interaction (PPI) network was build to identified hubs in the network. The authors declared that these hub genes would be used to build a prediction model. Further, proteomics data demonstrated that 4 hub genes are involved in the transformation process of HCC after HBV infection. However, the authors fail to provide some important details of their data analysis and an overall evaluation of their model, which is very important if clinicans would like to use their model to help the corresponding diagnosis. For that reason, I would suggest that the following is undertaken to strengthen the existing model and support the conclusions made, before considering the manuscript for potential publication:

  1. The aim of this project is identify the key genes involveld in HBV-Related Hepatocellular Carcinoma. In the manuscript, the key genes are selected and validated on the basis of the comparision between HCC and cirrhosis. The author need to explain the rational why they focus on the comparison between HCC and cirrhosis since the HCC is a complicated disease and so many distinct factors could trigger the pathogenesis of HCC in so many differenct perspective

Response 1:

We much appreciate your suggestions, which have given us more to consider.

We discovered that the content structure of the abstract and introduction is unclear and does not stress the article's major ideas, which may make it difficult for readers to understand. Therefore, we have revised this section extensively to improve the reading experience for audiences. Our work aims to discover critical genes involved in the progression from cirrhosis to liver cancer and to provide valuable research direction for the early detection of liver cancer in cirrhotic individuals. The identification of genes that are uniquely expressed in HBV-related HCC is merely a preliminary step in the objective of this study. On the basis of specifically expressed genes in HCC, we will search for critical genes implicated in the transition from cirrhosis to HCC. Consequently, the identification of genes that are uniquely expressed in HBV-related HCC is simply a preliminary step in achieving the purpose of this study.

The following are the primary justifications for selecting cirrhosis as the study phase:

The vast majority of HCC result from persistent liver inflammation and cirrhosis. Cirrhosis is the final stage of the disease before liver cancer develops and is one of the risk factors for the development of liver cancer. The incidence of HCC is much higher in the cirrhotic state than in the non-cirrhotic state. However, there are significant obstacles to primary care surveillance for HCC in cirrhotic patients. Little is known about practice patterns for HCC primary care surveillance, with less than 20% of patients with cirrhosis undergoing HCC testing every 6 months. And primary care testing for HCC is confounded, with many primary care providers (PCP) erroneously believing that clinical examination of alpha-fetoprotein (AFP) expression was effective surveillance tool. At a threshold of 20 ng/mL, the sensitivity of AFP for detecting HCC smaller than 5 cm ranges from 49% to 71%, while the specificity ranges from 49% to 86%. And AFP does not clearly differentiate between HCC and cirrhosis, which severely limits its utility for guiding individualized diagnostic and therapeutic strategies. Moreover, modern medical research demonstrates that the liver has no pain sensation, and even if liver disease has begun, the body is unable to perceive it through pain feedback mechanisms. During the transformation of cirrhosis to liver cancer, there are no obvious clinical symptoms, causing many patients to be in the middle and late stages of the disease when liver cancer is diagnosed, resulting in an exceedingly poor prognosis and increased patient mortality. The presence of both inflammation and cirrhosis complicates the early detection of HCC, and very few HCC biomarkers have adequate diagnostic sensitivity for HCC in its early stages. Even though patients are routinely monitored for HCC, diagnosis will be missed because of poor accuracy and sensitivity of surveillance methods .

However, the process by which cirrhosis transforms into liver cancer is still unknown. Current research on the pathogenesis of HCC focuses on identifying biomolecules involved in cancer progression and tumor cell response, particularly those specifically activated by HBx. HBx, the principal oncogenic protein of HBV, may perform a crucial function in hepatocarcinogenesis. Understanding HBx-mediated driver complex aspects of HCC, particularly HBx-regulated downstream genes, would facilitate the search for biomarkers from a carcinogenic perspective and construct an effective link between biomarkers and the disease process, thereby assigning "stage" symbolic attributes to markers offers a novel strategy for the effective prevention, diagnosis, and treatment of hepatocarcinogenesis in cirrhotic patients .

Point 2: The details of prediction model is very important to support the authors’ aim of providing more precision diagnosis of HCC on the gene levels. The details of the prediction model should be added to the revised version of this paper.

Response 2:

Thank you very much for your comments. We strongly concur your comments, and this will be the focus of our research in the following stage. We are collecting tissue samples and clinically relevant data from hospital patients in an attempt to validate the expression trends of these key Hub genes we identified in the transition from cirrhosis to HCC, and to further analyze their clinical diagnostic value for HCC. Therefore, in the following phase of work, we will share more specifics regarding the model's predictions.

The present study aims to identify the essential molecules involved in the development of cirrhosis to HCC. The assessment of the clinical value of these Hub genes based on database information at this stage is only an auxiliary way to prove it.

Point 3: The details of using the Nomogram to predict the overall survival is missing in this manuscript. I did see the authors used the package “R-forestplot” to creat a nomogram. Since Nomogram provide very important support that these hub genes could predict the survival of distinct patients, the details of buidling and using Nomogram should be further provided at least in the supplementary methods;

Response 3:

Thank you very much for your comments.

The following description of the Nomogram's construction has been added to the article's methodology:

“The objective of constructing a Cox proportional hazards regression model was to assess the relative contribution of key genes to patient survival prediction. We visualized the univariate Cox regression analysis using "R-forestplot" and "R-survival" package and created a nomogram based on the optimal multivariate Cox regression analysis to predict 1-year, 3-year, and 5-year survival probabilities using "R-survival", "R-rms" and "R-ggplot2” packages.”

Point 4: Traditionaly, C-index of Nomogram larger than 0.85 is considered the reliable prediction of survival probability. If the author insist that 0.702, which is lower than 0.85, is already good enough, the authors should demonstrate the corresponding evidences

Response 4:

Thank you very much for your comments.

The associated evidence is as follows:

PMID: 34805014, PMID: 28982688, PMID:35568377, PMID:31930042, PMID:23358969.

The above references are also cited in that section of the article to strengthen the credibility of the experimental results.

Point 5: The 120 DEGs and the hub genes are very important since this is the key discovery of this project. The authors should provide the list of these DEGs and hub genes in their manuscript;

Response 5:

Thank you very much for your comments.

This section has been added to the supplementary Tables; please refer to Tables S1 and S2 for details.

Table S1.  Exploration of 120 potential DEGs in the GSE121248 dataset.

Gene symbol

Gene description

logFC

P value

Up/down

LINC01419

Long Intergenic Non-Protein Coding RNA 1419

2.084786602

1.94E-06

Up

MAGEA6

MAGE Family Member A6

2.171022861

1.13E-05

Up

PEG10

Paternally Expressed 10

2.042842120

4.82E-05

Up

ACSL4

Acyl-CoA Synthetase Long Chain Family Member 4

2.813819514

2.85E-13

Up

GPC3

Glypican 3

3.847841892

1.89E-15

Up

CTHRC1

Collagen Triple Helix Repeat Containing 1

2.337200459

2.39E-09

Up

COL15A1

Collagen Type XV Alpha 1 Chain

2.968063583

2.40E-21

Up

SPP1

Secreted Phosphoprotein 1

2.101476266

5.48E-07

Up

SULT1C2

Sulfotransferase Family 1C Member 2

2.107128429

7.96E-08

Up

AKR1B10

Aldo-Keto Reductase Family 1 Member B10

3.068006046

5.05E-09

Up

RBM24

RNA Binding Motif Protein 24

2.111902112

5.85E-12

Up

CAP2

Regulatory Protein 2

2.470035042

9.68E-26

Up

HMMR

Hyaluronan Mediated Motility Receptor

2.220242695

1.27E-20

Up

PBK

PDZ Binding Kinase

2.217813595

5.08E-16

Up

RRM2

Ribonucleotide Reductase Regulatory Subunit M2

2.248686305

4.98E-18

Up

ECT2

Epithelial Cell Transforming 2

2.038996714

4.09E-19

Up

BUB1B

BUB1 Mitotic Checkpoint Serine/Threonine Kinase B

2.049502320

2.77E-17

Up

PRC1

Protein Regulator Of Cytokinesis 1

2.085868676

4.28E-19

Up

CCNB1

Cyclin B1

2.053859931

7.24E-17

Up

TOP2A

DNA Topoisomerase II Alpha

3.261963386

8.89E-24

Up

NEK2

NIMA Related Kinase 2

2.314657830

7.55E-18

Up

CDK1

Cyclin Dependent Kinase 1

2.029493985

2.54E-16

Up

ANLN

Anillin, Actin Binding Protein

2.499189772

6.93E-19

Up

DTL

Denticleless E3 Ubiquitin Protein Ligase Homolog

2.197331263

1.05E-17

Up

RACGAP1

Rac GTPase Activating Protein 1

2.002401328

2.12E-22

Up

ASPM

Assembly Factor For Spindle Microtubules

2.859644768

4.05E-22

Up

ZIC2

Zic Family Member 2

2.045716266

1.11E-10

Up

IGF2BP3

 Insulin Like Growth Factor 2 MRNA Binding Protein 3

2.238629923

3.59E-09

Up

CRNDE

Colorectal Neoplasia Differentially Expressed

2.652245521

8.79E-13

Up

ROBO1

Roundabout Guidance Receptor 1

2.024942977

5.94E-15

Up

SPINK1

Serine Peptidase Inhibitor Kazal Type 1

3.656277417

6.58E-10

Up

SLCO1B3

Solute Carrier Organic Anion Transporter Family Member 1B3

-2.927220730

6.87E-10

Down

ADH4

Alcohol Dehydrogenase 4 (Class II), Pi Polypeptide

-2.121819958

2.47E-06

Down

SLC22A1

Solute Carrier Family 22 Member 1

-2.529147583

2.49E-09

Down

ESR1

Estrogen Receptor 1

-2.275245143

1.02E-14

Down

FLJ22763

Chromosome 3 Open Reading Frame 85

-2.715307483

7.61E-15

Down

LINC00844

Long Intergenic Non-Protein Coding RNA 844

-2.349710946

8.90E-07

Down

C9

Complement C9

-2.786415328

1.46E-07

Down

FAM134B

Reticulophagy Regulator 1

-2.339624903

1.85E-11

Down

THRSP

Thyroid Hormone Responsive

-2.993124985

2.11E-09

Down

TENM1

Teneurin Transmembrane Protein 1

-2.375247127

2.96E-10

Down

FOLH1B

Folate Hydrolase 1B (Pseudogene)

-2.113008154

3.54E-14

Down

BBOX1

Gamma-Butyrobetaine Hydroxylase 1

-3.296616849

3.01E-15

Down

SRD5A2

Steroid 5 Alpha-Reductase 2

-2.015257394

8.89E-17

Down

ACSM3

Acyl-CoA Synthetase Medium Chain Family Member 3

-2.028393911

5.43E-14

Down

GBA3

Glucosylceramidase Beta 3 (Gene/Pseudogene)

-2.463556707

1.26E-10

Down

CYP2B6

Cytochrome P450 Family 2 Subfamily B Member 6

-2.434282425

7.19E-12

Down

GLYAT

 Glycine-N-Acyltransferase

-2.195290004

4.40E-10

Down

ZG16

Zymogen Granule Protein 16

-2.120752216

8.68E-17

Down

GLS2

Glutaminase 2

-2.400286201

3.28E-11

Down

MFSD2A

MFSD2 Lysolipid Transporter A, Lysophospholipid

-2.316298201

9.18E-14

Down

HAO2

Hydroxyacid Oxidase 2

-2.620033332

3.14E-12

Down

CYP2A6

Cytochrome P450 Family 2 Subfamily A Member 6

-2.322074035

4.79E-09

Down

CYP2A7

Cytochrome P450 Family 2 Subfamily A Member 7

-2.024761409

3.87E-11

Down

GYS2

Glycogen Synthase 2

-2.291074305

2.76E-08

Down

C3P1

Complement Component 3 Precursor Pseudogene

-2.088864100

4.97E-10

Down

LPA

Lipoprotein(A)

-2.368064154

2.20E-14

Down

AKR1D1

 Aldo-Keto Reductase Family 1 Member D1

-2.557761216

4.72E-09

Down

GHR

Growth Hormone Receptor

-2.282112510

4.99E-11

Down

BCHE

Butyrylcholinesterase

-2.275286332

1.24E-08

Down

SULT1E1

Sulfotransferase Family 1E Member 1

-2.116352834

1.41E-12

Down

CNTN3

Contactin 3

-2.416032942

1.01E-08

Down

VNN1

Vanin 1

-2.262903436

2.18E-08

Down

CLRN3

Clarin 3

-3.124813776

2.61E-16

Down

CYP39A1

Cytochrome P450 Family 39 Subfamily A Member 1

-2.651808560

1.73E-12

Down

FREM2

FRAS1 Related Extracellular Matrix 2

-2.369084568

1.58E-11

Down

IGF1

Insulin Like Growth Factor 1

-2.348748467

4.12E-13

Down

MT1M

Metallothionein 1M

-2.834980961

4.68E-10

Down

CXCL2

C-X-C Motif Chemokine Ligand 2

-2.065043876

1.63E-12

Down

CD5L

CD5 Molecule Like

-2.054764444

1.33E-16

Down

FAM65C

RIPOR Family Member 3

-2.459200521

2.76E-22

Down

CRHBP

Corticotropin Releasing Hormone Binding Protein

-2.999371683

1.19E-24

Down

CLEC1B

C-Type Lectin Domain Family 1 Member B

-3.135877707

7.65E-30

Down

FCN3

Ficolin 3

-3.341982552

3.35E-23

Down

OIT3

Oncoprotein Induced Transcript 3

-3.229522436

1.19E-26

Down

PLAC8

Placenta Associated 8

-2.065061375

1.78E-20

Down

CXCL14

C-X-C Motif Chemokine Ligand 14

-3.978631486

1.53E-32

Down

HHIP

Hedgehog Interacting Protein

-2.803161676

1.78E-28

Down

CXCL12

C-X-C Motif Chemokine Ligand 12

-2.786652251

6.57E-17

Down

LIFR

LIF Receptor Subunit Alpha

-2.400421259

1.32E-18

Down

HAMP

Hepcidin Antimicrobial Peptide

-4.160273583

9.72E-18

Down

HGFAC

HGF Activator

-2.827246907

4.24E-18

Down

CDHR2

Cadherin Related Family Member 2

-2.446512749

2.71E-28

Down

IGFBP3

Insulin Like Growth Factor Binding Protein 3

-2.075588382

3.11E-19

Down

FOSB

FosB Proto-Oncogene, AP-1 Transcription Factor Subunit

-2.429478842

5.44E-15

Down

FOS

Fos Proto-Oncogene, AP-1 Transcription Factor Subunit

-2.191916691

5.23E-14

Down

PGLYRP2

Peptidoglycan Recognition Protein 2

-2.162003459

5.42E-11

Down

APOF

Apolipoprotein F

-2.685010309

5.08E-14

Down

KMO

Kynurenine 3-Monooxygenase

-2.094420506

1.05E-14

Down

BCO2

Beta-Carotene Oxygenase 2

-2.333059189

3.39E-15

Down

CYP1A2

Cytochrome P450 Family 1 Subfamily A Member 2

-3.353119355

2.03E-19

Down

NAT2

N-Acetyltransferase 2

-2.481342467

2.86E-17

Down

DNASE1L3

Deoxyribonuclease 1 Like 3

-2.210444745

4.13E-15

Down

LCAT

Lecithin-Cholesterol Acyltransferase

-2.519286270

8.29E-23

Down

ZGPAT

Zinc Finger CCCH-Type And G-Patch Domain Containing

-2.195422544

1.05E-25

Down

KCNN2

Potassium Calcium-Activated Channel Subfamily N Member 2

-3.990836923

9.48E-28

Down

LINC01093

Long Intergenic Non-Protein Coding RNA 1093

-4.178427405

1.27E-24

Down

IDO2

Indoleamine 2,3-Dioxygenase 2

-3.321683185

1.20E-21

Down

IGFALS

Insulin Like Growth Factor Binding Protein Acid Labile Subunit

-2.047566911

6.13E-31

Down

CLEC4M

C-Type Lectin Domain Family 4 Member M

-2.871704973

1.60E-26

Down

FCN2

Ficolin 2

-3.636464429

2.08E-26

Down

CLEC4G

C-Type Lectin Domain Family 4 Member G

-3.162168263

6.20E-26

Down

CYP26A1

Cytochrome P450 Family 26 Subfamily A Member 1

-2.721213193

1.02E-24

Down

CNDP1

Carnosine Dipeptidase 1

-3.752327730

1.09E-22

Down

TTC36

Tetratricopeptide Repeat Domain 36

-2.957724000

3.21E-20

Down

TMEM27

Collectrin, Amino Acid Transport Regulator

-3.412451340

3.44E-20

Down

SLC25A47

Solute Carrier Family 25 Member 47

-2.179932568

3.05E-18

Down

IGHM

Immunoglobulin Heavy Constant Mu

-2.288931297

8.38E-17

Down

IGLC1

Immunoglobulin Lambda Constant 1

-2.299163764

2.05E-10

Down

IGLV1-44

Immunoglobulin Lambda Variable 1-44

-2.123044363

3.52E-10

Down

TACSTD2

Tumor Associated Calcium Signal Transducer 2

-2.189690710

1.42E-10

Down

CDH19

Cadherin 19

-2.201681135

8.57E-16

Down

PDGFRA

Platelet Derived Growth Factor Receptor Alpha

-2.360296332

1.63E-10

Down

C7

Complement C7

-2.881514054

6.25E-13

Down

DCN

Decorin

-2.512236660

1.42E-11

Down

SRPX

Sushi Repeat Containing Protein X-Linked

-2.808193819

2.90E-18

Down

HGF

Hepatocyte Growth Factor

-2.710907042

3.06E-17

Down

WDR72

WD Repeat Domain 72

-2.008030745

7.80E-09

Down

KANK4

KN Motif And Ankyrin Repeat Domains 4

-2.230789907

9.14E-17

Down

ENO3

Enolase 3

-2.179728417

1.64E-17

Down

Table S2. The top 20 Hub genes identified by the plugin CytoHubba in Cytoscape software.

Gene symbol

Gene description

Degree

SPP1

Secreted Phosphoprotein 1

9

HMMR

Hyaluronan Mediated Motility Receptor

13

PBK

PDZ Binding Kinase

13

RRM2

Ribonucleotide Reductase Regulatory Subunit M2

13

CCNB1

Cyclin B1

15

CDK1

Cyclin Dependent Kinase 1

15

ASPM

Assembly Factor For Spindle Microtubules

15

DTL

Denticleless E3 Ubiquitin Protein Ligase Homolog

13

NEK2

NIMA Related Kinase 2

13

ECT2

Epithelial Cell Transforming 2

13

RACGAP1

Rac GTPase Activating Protein 1

13

PRC1

Protein Regulator Of Cytokinesis 1

13

TOP2A

DNA Topoisomerase II Alpha

14

ANLN

Anillin, Actin Binding Protein

13

BUB1B

BUB1 Mitotic Checkpoint Serine/Threonine Kinase B

13

HGF

Hepatocyte Growth Factor

9

CXCL12

C-X-C Motif Chemokine Ligand 12

9

DCN

Decorin

7

ESR1

Estrogen Receptor 1

15

IGF1

Insulin Like Growth Factor 1

12

Point 6: The authors mentioned that their genomics and proteomics data demonstrated great promise for the early diagnosis of HCC. If no direct evidence of better earlier diagnosis of HCC would be shown in the manuscript, I am afraid that the author should not make this narrative in their manuscript

Response 6:

Thank you very much for your comments.

The description of the original article appears below.

“Advances in genomics and proteomics for the detection of genetic biomarkers have demonstrated great promise for the early diagnosis of HCC.” We only like to emphasize the advancements in detection technologies.

Point 7: Both PCA and UMAP is used in this project, since not all the readers of Genes are statistician, the authors should demonstrate their rational of using both methods;

Response 7:

Thank you very much for your comments. We strongly agree with you, so the reasons for using PCA and UAMP analysis have been stated in the article. as follows,

“Using PCA and UMAP, only two clusters were identified, and the accuracy of the groups’ characterization was confirmed by observing that most patients within the group were assigned to the same cluster”.

“Prior to this, we used the clustering techniques PCA and UAMP to ensure that there were no clusters with similar intrinsic characteristics between HBV-infected HCC tissues group and para-cancer liver tissues group, which could improve the credibility of the grouping [31]”.

Point 8: In figure 3, the author shows the results of KEGG in their panel C and D, they should add more details such as what is the differences between the blue dots and red dots in their Figure Legend

Response 8:

Thank you very much for your comments. To improve the reading experience, we replaced the bubble chart with a bar chart and added more information.

Figure 3. GO and KEGG enrichment analysis of differentially expressed genes in HBV-related HCC. GO term enrichment results (A, B). KEGG enrichment results (C, D). The blue nodes represent the entries, the red nodes represent the numerators, and the connecting lines represent their relationship.

Point 9: This manuscript have some typos, such as in line 347, “DGEs” should be “DEGs”, further proofreading is needed;

Response 9:

Thank you very much for your comments.

We have changed “DGEs” to “DEGs”. At the same time, we have further proofread the content of the article.

Point 10: The English of this manuscript still has space to improve. If possible, the authors should find a native speaker to proofread the whole manuscript.

Response 10:

We invited the native English speaker to help language polishing, and hope our improvements of language could meet the reviewers’expectation.

Reviewer 3 Report

Authors mentioned "Our team identified potential Hub genes involved in HBV-associated HCC and also used proteomic analysis to demonstrate for the first time the role of these Hub genes in hepatocarcinogenesis at the level of oncogenicity." Unfortunately, I'm not agreed with this statement.

I feel it will be great if authors will present some downstream signaling pathways. They mentioned those but no functional analysis. In my opinion Functional analysis of those predicted downstream pathways can make this manuscript scientifically sound.

Author Response

Point 1: Authors mentioned "Our team identified potential Hub genes involved in HBV-associated HCC and also used proteomic analysis to demonstrate for the first time the role of these Hub genes in hepatocarcinogenesis at the level of oncogenicity." Unfortunately, I'm not agreed with this statement.

Response 1:

Thank you very much for your comments and constructive suggestions. We will be more stringent in our manuscript editing, and the following corrections have been made to the article. “Our team identified potential diagnostic biomarkers involved in the progression of cirrhosis to HCC from a carcinogenic perspective.”

Point 2: I feel it will be great if authors will present some downstream signaling pathways. They mentioned those but no functional analysis. In my opinion Functional analysis of those predicted downstream pathways can make this manuscript scientifically sound.

Response 2:

Thank you very much for your comments and constructive suggestions.

We are thrilled that our views align so closely with yours.

Our team has been exploring the functional analysis of these key Hub genes involved in the transformation of cirrhosis to HCC.

We have proven the role of RRM2 in the proliferation of HCC and this finding has been reported in our “in press” article entitled “A pan-cancer analysis of the oncogenic role of ribonucleotide reductase subunit M2 in human tumors”, as shown in the figure below.

We verified the upregulated protein expression of RRM2 in tumor tissue compared to adjacent normal tissue by IHC staining (Fig.A, B). Edu assays (Fig. F-H) showed that inhibiting RRM2 significantly reduced the proliferation rate of HepG2 and Huh-7 cells. In addition, we found that RRM2 mRNA expression were significantly upregulated in LO2-HBx cells (Fig. J), confirming that RRM2 serves as an oncogene in HCC.

Figure Validation of the oncogenic role of RRM2 in HCC.

(A, B) Comparison of RRM2 expression level between tumor and adjacent normal tissues in HCC cohort (n=154). Scale bar: 200 μm. Non-targeting (negative control) or siRNA targeting RRM2, as indicated. After transfection, RNA samples were then collected and relative expression ratios of RRM2 in HepG2 cells (C) and Huh-7 cells (D) were determined using qPCR analysis (n=3). (E) CCK-8 was used at 48h post-transfection to evaluate the viability of Huh-7 and HepG2 cells (n=3). (F) The cell proliferation was examined by Edu incorporation assay. Scale bar:100μm. (G, H) Quantification of positive cells in the Edu assay (n=3). The percentage of positive cells was determined by counting 1000 cells/sample. the HA-HBx construct or empty vector was transfected into LO2 cells for 24h. PCR analysis of HBx (I) and RRM2 (J) mRNA expression in LO2 cells with above treatment (n=3). The differences between two groups were estimated by the Student’s t test. All values are the mean±SD. * P <0 .05, ** P <0 .01, ***P <0 .001, **** P <0 .0001.

As for the functional investigation of other Hub genes, it will be conducted in stages, and preparations are already being made.
